# Protection of Mitochondria, Cells and Organs from Ischemia–Reperfusion Damage Through Preventive Redox Bioregulation by Ozone

**DOI:** 10.3390/ijms26125557

**Published:** 2025-06-10

**Authors:** Renate Viebahn-Haensler, Olga Sonia León Fernández

**Affiliations:** 1Medical Society for the Use of Ozone in Prevention and Therapy, Iffezheim, D-76473 Baden-Baden, Germany; 2Pharmacy and Food Institute, University of Havana, Calle 222 # 2317 e/23 y 31, Coronela, Lisa, Havana 10 400, Cuba

**Keywords:** redox balance, oxidative stress, low-dose ozone, redox bioregulation, ischemia–reperfusion damage, mitochondrial biogenesis, ozone therapy

## Abstract

Ischemia–reperfusion (I/R) damage remains a major problem in surgery, primarily based on high oxidative stress generated during the reperfusion process. Mitochondria are significantly affected, their metabolic and energetic processes are impaired, and the redox system is out of balance. Regulation and restoration of the redox balance by oxidative preconditioning with ozone is being investigated worldwide in cell and animal models. Selected preclinical trials and their results, with a focus on cardiological and neuronal I/R damage, are presented and discussed. We regularly find an upregulation of antioxidants, demonstrated in SOD (superoxide dismutase) and GSH (glutathione, reduced form, and a decrease in oxidative stress as a result, shown here using the typical stress parameters, MDA (malondialdehyde) and TBARS (thiobarbituric acid reactive substances). Mitochondrial biogenesis, comparable to moderate physical activity, is induced by ozone oxidative preconditioning in an I/R model in rats and reviewed in this paper.

## 1. Introduction

Great expectations can be placed on secondary prevention during surgical interventions to prevent or minimize ischemia–reperfusion (I/R) injury through oxidative preconditioning by applying low-dose medical ozone: High oxidative stress is significantly reduced, the redox balance is maintained or largely restored: Mitochondria remain intact and can fulfill their metabolic and energetic obligations.

I/R damage due to a sudden oxygen flow during reperfusion is a major problem in surgery, e.g., in cardiology (heart attack), neurology (stroke), vascular surgery and, last but not least, in transplant surgery [1]. This results in an excess of reactive oxygen species (ROS) such as hydrogen peroxide, superoxide radicals, and hydroxyl radicals, which, together with oxygen, form lipoperoxides (LPO) and initiate radical chain reactions. Mitochondria are damaged and significantly lose their metabolic functions.

Excessive oxidative stress caused by I/R procedures, especially in coronary and cerebral surgery, requires antioxidant treatment strategies (vitamin C, vitamin E, Curcumin, N-acetylcysteine, melatonin, etc.), as shown by a large number of animal studies. As monotherapies, the results leave much to be desired and the search is on for suitable combinations and complementary concepts. The number of corresponding publications has increased considerably in recent years, as Figure 1 shows for the last 5 years.

In numerous preclinical studies, preventive ozone applications have achieved excellent results and open up a new concept of oxidative preconditioning with medical ozone according to the low-dose concept for optimal protection against I/R damage [2,3,4,5,6,7,8,9,10,11,12] and others.

## 2. I/R Damage and the Regulation by Oxidative Preconditioning Through Low-Dose Medical Ozone

Our focus here is on myocardial and cerebral I/R damage, which is probably the most common type that occurs after a heart attack or apoplexy. Fortunately, there are an increasing number of preclinical studies with promising results (12), which will certainly soon be followed by clinical studies. Here we refer only to those preclinical trials, in which typical redox parameters were measured to underline the redox regulation by ozone, and for the sake of clarity, significant, reliable, and clearly reproducible parameters were used, despite their small number: They are viewed either from the oxidative stress side and from the antioxidant side of the redox balance (see Table 1 and Figure 2).

Fundamental results of redox bioregulation by ozone are manifested in a regulation of the redox balance by ozone preconditioning to protect against I/R damage in surgery; shown in Figure 2 as a selection of eight preclinical studies in cell and animal models in which meaningful parameters were measured. We regularly find an upregulation of antioxidants, demonstrated here in SOD and GSH, and a decrease in oxidative stress as a result, shown here using the typical stress parameters, MDA and TBARS. In 1998, the León research group founded this idea, and it is now of worldwide interest, with good prospects for establishing a method that is very easy to handle, and due to its mechanism of action, it enables general application, e.g., in brain and heart surgery, in order to significantly reduce I/R damage [2]. Clinical studies should follow and are in progress. A short survey on the mechanism of action is displayed in Figure 3.

The values measured for the I/R groups are taken as baseline and the antioxidants represent the increase (green) after ozone preconditioning while the oxidative stress is characterized by a decrease (orange). Each bar represents the prevention of ROS induced injury in different organs as known from ischemia during surgery and subsequent reperfusion. Upregulating the cellular and mitochondrial antioxidants seems to be the main mechanism of systemically administered ozone at low concentrations and doses: oxidative stress decreases and the redox balance is restored.
Figure 2Ischemia–reperfusion injury and its protection by regulation of the redox balance through ozone preconditioning. Overview of the repair parameter (SOD or GSH) versus oxidative stress parameter (MDA or TBARS) in preclinical studies, as described in Table 1 [2,4,6,10,14,15,16,17].
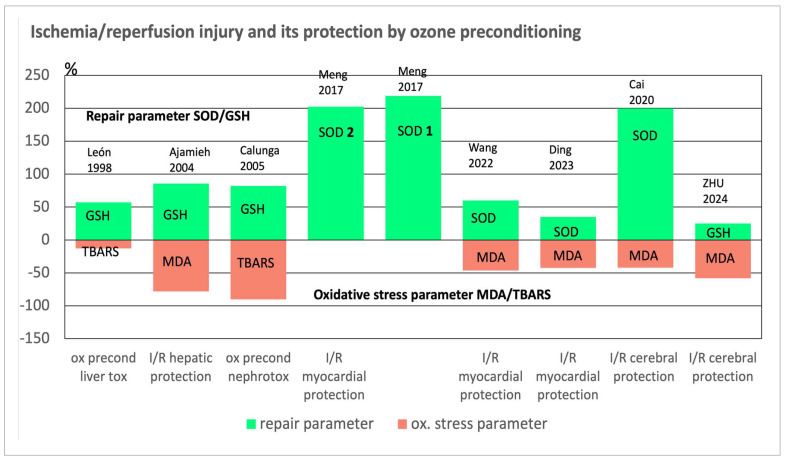


### 2.1. Mechanism of Action and the Ozone Effect

To start a surgical procedure, the blood flow is interrupted, generating ischemia over a certain period of time, at the end of which reperfusion is started, whereby an excess of oxygen floods ischemic areas not being metabolized by the overstrained mitochondria. Reactive oxygen compounds are formed, such as superoxide radicals, hydrogen peroxide, OH radicals, lipoperoxides, etc., with several cell and organ destructions as a consequence.

Ozone does not pass through the cell membrane (polar molecular structure), as is often assumed and repeatedly reported, nor does it decompose into oxygen and oxygen radicals; it has an extremely short half-life in blood and biological fluids and does not react directly with cell organelles or compartments. It follows an indirect reaction mechanism, interacting with signaling pathways (such as the Nrf2 or NFkB pathways), as is the case with several other drugs [18]. Ozone immediately reacts with isolated double bonds (cell membranes) to form short chain “ozone peroxides”, which are reduced by the glutathione system (GSH/GSSG) and pass on the information to the cell nucleus via the NFkB and Nrf2 pathway; finally, antioxidants are produced and the redox balance is restored, as described in detail in [19] (see Figure 3).
Figure 3Ischemia–reperfusion: oxidative stress and regulation by medical ozone. Ozone immediately reacts with isolated double bonds (cell membranes) to form short-chain “ozone peroxides”, which are reduced by the glutathione system (GSH/GSSG) and pass on the information to the cell nucleus via the NFkB and Nrf2 pathway; finally, antioxidants are produced. The number of reactive oxygen species (ROS) decreases, and the redox system can be balanced. GSHox: Glutathione peroxidase, GSred: Glutathione reductase; GGT: γ- glutamyl transferase, CAT: Catalase, SOD: Superoxide dismutase, NFkB and Nrf2: Nuclear factors, MDA: malondialdehyde [19,20,21].
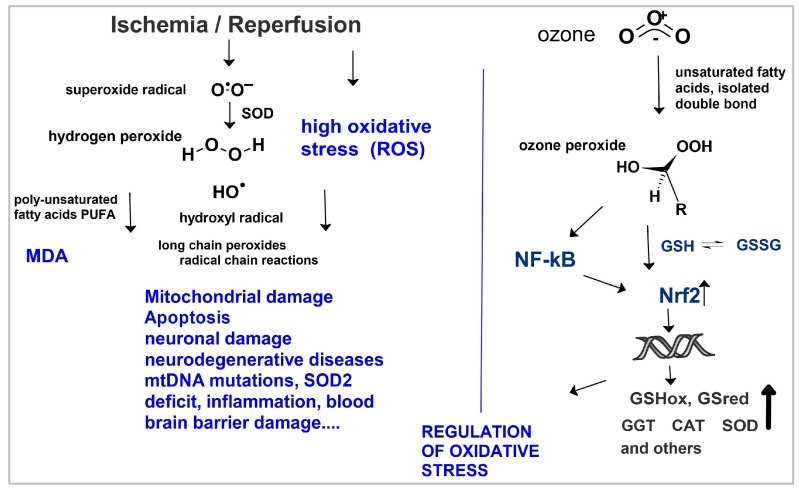


### 2.2. Heart Protection Against I/R Damage Through Ozone-Mediated Mitochondrial Biogenesis

Mitochondrial biogenesis: To maintain their specific functions, all organ cells (except RBCs) depend on intact mitochondria as energy suppliers. Perfect regulatory processes take place between the cells and their mitochondria: If the cell has a high energy requirement, mitochondrial activity increases, the metabolic rate intensifies, and ATP production improves. Moderate physical activity, such as playing sports, produces low and moderate oxidative stress, which leads to optimal intercommunication and signal transmission, and thus an improved oxygen supply. The mitochondria are also increasingly stimulated to multiply, i.e., divide (fission), when energy requirements increase, and the mitochondrial density in the cells increases as a consequence. Mitochondrial fusion, in balance with mitochondrial fission, serves, among other things, to repair defective or insufficient mitochondria.

This means that the number of mitochondria in an organ cell adapts to the energy requirement, thus improving cell protection and reducing the risk of oxidative stress.

In organs with a high energy requirement such as muscle cells, hepatocytes or the myocardial cells, there are from 1000 to several 100,000 (egg cells) mitochondria with a volume share of up to 25% of the cell volume [22].

The impaired regulation of the redox balance in the mitochondria or deficit of mitochondria determines the fate of the cell and results in a variety of diseases, including neurodegenerative diseases, chronic inflammations such as rheumatoid arthritis, vascular diseases, insulin resistance and diabetes type 2, aging and age-related diseases, and cancer, all of them accompanied by high oxidative stress [22,23]. This is already basic knowledge in mitochondria diseases; however, the role of ozone in a similar process remains unknown, and thus is discussed here.

The regulation of oxidative stress by ozone in ROS-induced mitochondriopathies was demonstrated earlier in cases of Rheumatoid arthritis, osteoarthritis, diabetes type 2 and aging described in more detail elsewhere [24].

Can Medical Ozone Induce Mitochondrial Fission and Increase the Mitochondrial Density, Similar to Moderate Sport Activity, with Its Mild Oxidative Stress Through Preventive Intraperitoneal Ozone Administration?

This chapter of the present paper focuses on oxidative preconditioning by ozone for the prevention of myocardial reperfusion injury in animal models. In long-lasting ischemia (>15 min) severe myocardial damage can occur followed by a broad spectrum of corresponding problems, such as destruction of myocytes due to muscle paralysis, vascular damage and irreversible cell damage induced by the enormous increase in reactive oxygen species (ROS) during reperfusion [25].

Surprisingly, histological studies revealed an increase in mitochondrial density after preventive intraperitoneal ozone administration according to the low-dose concept. However, this study attracted little interest and was probably forgotten due to language problems, but is highly relevant today with the increasing importance of mitochondrial medicine [13].

Animal model. In several preclinical studies oxidative preconditioning with ozone proved to be an effective method of regulating redox balance in order to prevent reperfusion damage (see Table 1).

On this basis, the effectiveness of ozone was tested in an ischemia–reperfusion model of the myocardium in animals: Thirty male albino rats weighing 100–150 g were divided into 3 groups, each with n = 10: group 1 served as control without pretreatment, these animals had to undergo only surgery, in order to eliminate that influence (sham operated). The animals in group 2 received a preventive i.p. ozone application 2x per week for 2 months and group 3 for 3 months (ozone concentration: 4 μg/mL; dose 28 μg per 100g rat corresponding to 400 μg per 100 mL blood), followed by I/R injury by 30 min ischemia and 30 min reperfusion. For detailed information, please refer to the original publication [13].

Result. After 2 months oxidative preconditioning with ozone, myocytes after I/R injury showed an increase in mitochondrial density by 33.9% compared to control (*p* < 0.05), while mitochondrial density decreased again to 16.8% after 3 months of preventive ozone administration compared to control (*p* < 0.05), see Figure 4.

A similar effect is known from moderate physical activity, e.g., in cardiac and skeletal muscle cells: the increased energy demand boosts mitochondrial activity combined with an increase in fission and autophagy processes. It can even compensate for the loss of mitochondria due to obesity as shown in an animal trial [26].

The induction of mitochondrial biogenesis by ozone application was first described in the literature by Barakat 2006 [13] as shown here.

The main contribution of ozone seems to be the interaction within the redox balance: maintaining or restoring the balance protecting the cells and organelles from dysfunctions by an excess of reactive oxygen species. Mitochondria can continue to fulfill their cellular obligations in metabolic regulation. Though the oxidative stressor MDA remains still high compared to the control, but the antioxidant capacity in the tissue increased.

The redox regulatory effect and mechanism of low-dose ozone are well known, frequently discussed and summarized in Figure 3 [2,19,20].

### 2.3. Protection of Mitochondria in the Heart Muscle Against I/R Damage

Coronary artery bypass surgery is frequently associated with I/R damage, which can be significantly reduced by oxidative preconditioning through i.p. ozone application at low doses, shown here in an animal model [14].

Procedure: Thirty Adult male Sprague-Dawley rats (200–250 g) were used, with five groups (sham group, I/R group, ozone + I/R and two control groups) each containing six animals; the “ozone group” animals received 100 μg/kg daily for five days (e.g., 25 μg per rat in 2 mL volume) preventing the I/R process (30 min of ischemia followed by 2 h of reperfusion). Among the critical protection parameters, the antioxidative enzymes SOD 1 (Cu, Zn SOD) and SOD 2, the manganese-dependent SOD (Mn-SOD) in the mitochondria increased in a statistically significant manner with *p* < 0.05 compared to the I/R animal group (see Figure 2).

The transcription factor Nrf2 induces this process of corresponding protein production in order to ultimately control the redox balance. We must be aware that Nrf2 itself is a crucial factor that must be kept in a balance, in order not to initiate metastasis in cancer patients. This means that ozone concentrations and doses should follow the low-dose concept [27,28].

In this study, Nrf2 mRNA decreased by 62% as a result of the I/R process compared to the sham-operated (control) group, whereas ozone oxidative preconditioning considerably attenuated this process (ozone + I/R group): Nrf2 was only 11% below the sham-operated group. The mitochondria remained largely intact, and only mild myocardial apoptosis was observed [14].

Protection of the myocardium from I/R injury by preventive low-dose ozone application (20 μg/mL here; 2 mL per animal)) is reversed by blocking Nrf2 and its pathway, as shown in cell and animal models. Oxidative stress and repair parameters behave accordingly (see Figure 2). As a result of restoring the redox balance, mitochondria damage and feroptosis (Fe-dependent apoptosis) is kept low and the infarct size small [17]. Good results are also reported at higher ozone concentrations but oxidative stress remained at a high level and could only be downregulated by drastically reducing the ozone concentration, while maintaining the dose (see Table 1 and Figure 2) [16]. These results are fully consistent with other studies on oxidative preconditioning with ozone: redox bioregulation, upregulation of cellular antioxidants, and the corresponding downregulation of oxidative stress, see Figure 2. Since myocardial infarction in particular is a very common and steadily increasing health problem, these results are promising and, as a preclinical study, provide a solid basis for further research, especially in clinical studies [27,28].

## 3. Protection Against Brain I/R Injury

### 3.1. Cell Model: Protection of Neuronal Cells

The SH-SY5Y cell line is widely used in in vitro trials to study neuronal viability and functions as a basis for neurodegenerative diseases [28].

As an ozone concentration of 20 μg/mL showed the best viability and activity of the neuronal cells compared to 40 μg/mL and 80 μg/mL; this was the concentration of choice for ozone oxidative preconditioning in this I/R cell model in order to protect the mitochondria from losing their functions and the cells from damage and apoptosis.

Again, stress parameters were reduced, with MDA representing oxidative stress by 42% and SOD representing the antioxidants as repair enzymes, increased by 200% compared to the I/R group and shown in Figure 2. The redox balance shifts towards the norm, mitochondria are protected, cytochrome C-release as a measure for permeability is attenuated, apoptosis increased by the I/R process to 37.1% of the cells, whereas ozone preconditioning decreased it to 28.4%. Control: apoptosis in 13% of the cells [15]. These results demonstrate protection against cerebral I/R injury by ozone at the cellular level as a first step, which is necessary before starting an animal study, and show that the basic idea of the ozone effect can be tested and confirmed using an animal model.

### 3.2. Cerebral I/R Injury Animal Model

The protective effect of ozone in I/R injury by maintaining the redox balance of the cell, protecting mitochondrial damage and reducing cell apoptosis (here ferroptosis as iron-dependent apoptosis) was demonstrated in an in vitro model using SH-SY5Y cells, a cell line derived from human bone marrow. As the cerebral I/R injury is usually followed by neuronal death or loss with all the well-known consequences; ozone preconditioning, with its protective effect, opens up new aspects and treatment strategies for cerebral I/R injuries and neuronal disorders in prevention and therapy.

Middle cerebral artery occlusion/reperfusion (MCAO/R) as a standard animal model is widely used to gain basic knowledge on focal cerebral ischemia in humans.

In the procedure, 112 rats (260–300 g) were randomly divided into seven groups, 16 animals in each group. We compared only three groups to emphasize the ozone effect and ozone mechanism: sham group as control, MCAO-group: middle artery occlusion for 120 min, followed by surgery and reperfusion. Ozone + MCAO: intraperitoneal pretreatment with ozone (oxidative preconditioning) before MCAO; ozone concentration: 40 μg/mL with 2,5 mL/kg/d for 5 days.

Result. Ozone preconditioning protected the brain from cerebral reperfusion injury and neuronal damage, and the infarction area was significantly reduced compared to the MCAO-group. By activating the Nrf2 pathway, ozone preconditioning leads to a powerful antioxidant capacity reducing the critical reactive oxidant species and, among other factors, crucial parameters, such as LPO, MDA, and GSH, in rat brain are kept close to the healthy range, as displayed in Figure 5. The infarct area (by 62.3%) is reduced, and the number of intact neurons compared to the MCAO-group showing an increase by 83.2% [28]. This preclinical study should be transferred to a clinical study in which the redox–bioregulatory effect of ozone is used for secondary prevention to minimize brain damage and neuronal loss. The classic applications of ozone are used in humans: MAH (major autohemo-therapy by enriching 50–100 mL of blood extracorporeally with 50–100 mL of ozone) or rectal ozone insufflation at low ozone concentrations are the methods of choice, described and discussed in guidelines [29].

## 4. Conclusions and Future Perspectives

Ischemia–reperfusion damage remains a major problem in surgery, cardiology (heart attack), neurology (stroke), vascular surgery and, last but not least, in transplant surgery, primarily based on high oxidative stress generated during the reperfusion process. At low concentrations and low doses, ozone oxidative preconditioning provides a new treatment concept to restore the redox balance, to protect the mitochondria and maintain the cell integrity. Basic mechanisms, treatment procedure concentrations and doses in the non-toxic ranges are well known, and guidelines have been published [29]. This knowledge, gained in a series of preliminary studies in different research groups with the same results, is presented here in myocardium and brain I/R ischemia–reperfusion models. This basic research could and should be transferred directly into clinical studies; in view of the large number of patients with infarcts, especially heart attacks and cerebral infarcts, several studies should be carried out to transfer this low side-effect method of the secondary prevention of high oxidative stress and its pathological sequelae into the clinic and, if possible, establish it as a standard. This opens up a broad spectrum for the integration of medical ozone as a protective measure in surgery.

## Figures and Tables

**Figure 1 ijms-26-05557-f001:**
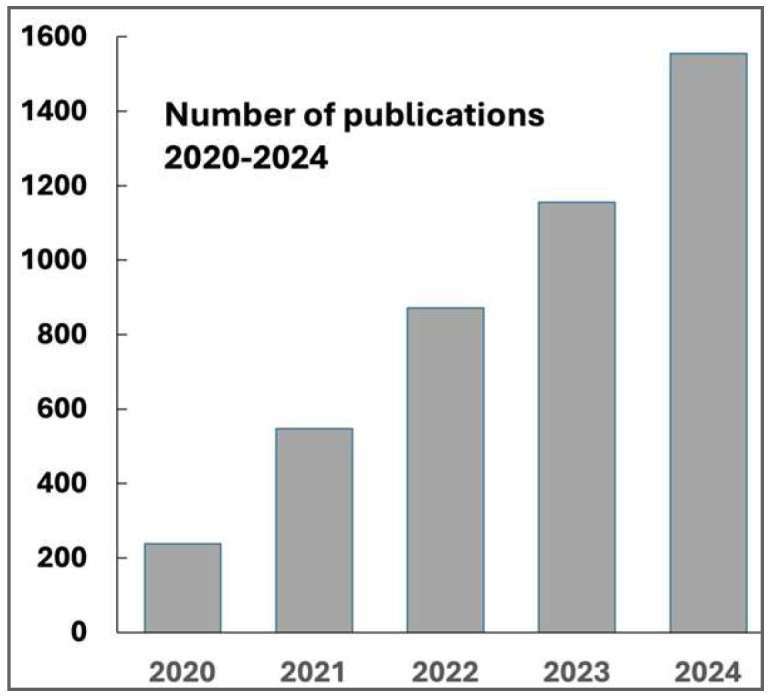
Treatment with antioxidants in I/R damage (myocard infarction and stroke); number of publications in 2020–2024 (database: PubMed).

**Figure 4 ijms-26-05557-f004:**
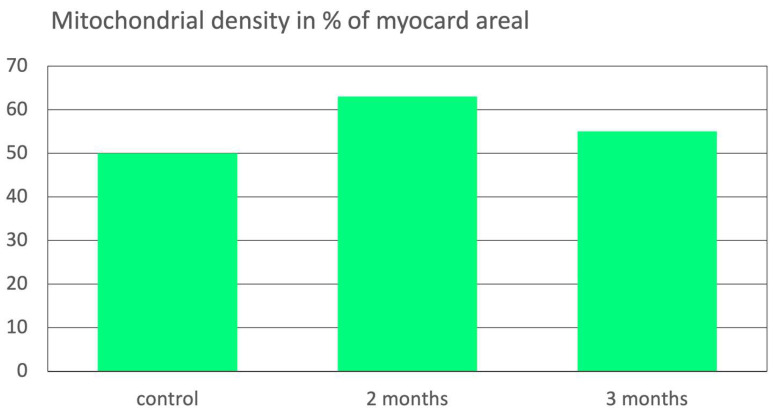
Mitochondrial density as a measure for mitochondrial fission (biogenesis) in the myocardium after oxidative preconditioning by ozone during 2 and 3 months. The number of mitochondria plays a special role for the myocardium with its high energy requirements. Mitochondrial density measured as % of the myocard areal in an ischemia–reperfusion model in animals. Each group with 10 animals (rats). Control: no pretreatment, Ischemia 30 min, reperfusion 30 min. “2 months”: these 10 rats were pretreated with intraperiteneal (i.p.) ozone applications for 2 months twice per week. “3 months”: same procedure as the “2 months group” for 3 months. Ozone concentration: 4 μg/mL; dose 28 μg per 100 g rat corresponding to 400 μg per 100 mL blood, followed by I/R injury by 30 min ischemia and 30 min reperfusion [13].

**Figure 5 ijms-26-05557-f005:**
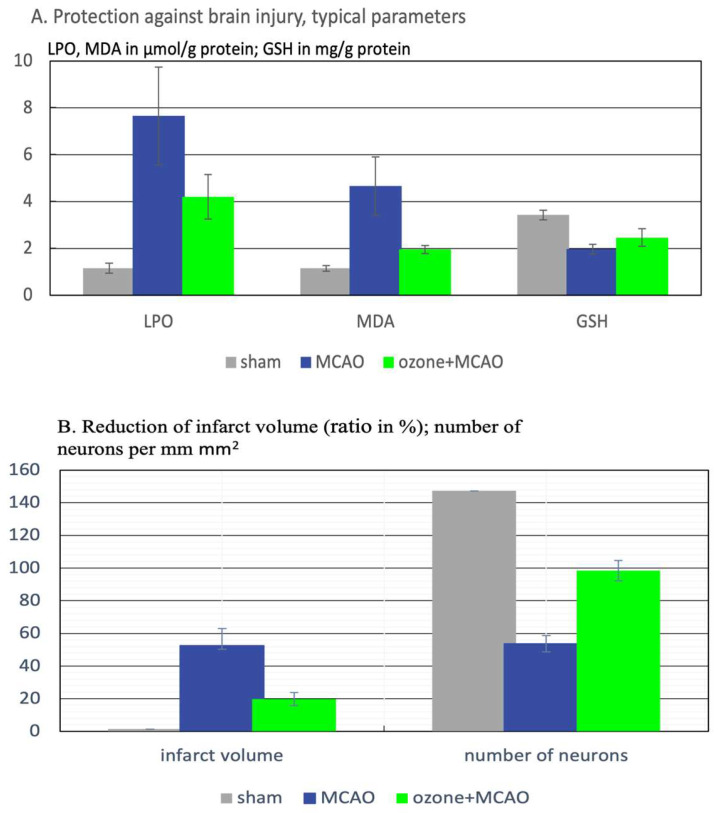
Protection against brain injury in an ischemia–reperfusion model in rats. (**A**) Critical parameters backing the underlying ozone mechanism. LPO, MDA in μmol/g protein; GSH in mg/g protein. All values are mean ± SD. Statistical results, LPO: sham vs. MCAO, *p* < 0.001; MCAO vs. ozone + MCAO, *p* < 0.01. MDA: sham vs. MCAO, *p* < 0.0001; MCAO vs. ozone + MCAO, *p* < 0.001. GSH: sham vs. MCAO, *p* < 0.0001, MCAO vs. ozone + MCAO, *p* < 0.05. (**B**) Reduction in infarct volume (ratio in %); number of neurons per mm^2^. Statistical results, Infarct volume: (sham = O) MCAO vs. ozone + MCAO, *p* < 0.1. number of neurons: sham vs. MCAO, *p* < 0.0001; MCAO vs. ozone + MCAO, *p* < 0.0001 [10].

**Table 1 ijms-26-05557-t001:** Preclinical studies (12) on the prevention of reperfusion injuries and intoxication by ozone oxidative preconditioning (without any claim to completeness).

Subject, Type of Study	Procedure	Antioxidants/Oxidative Stress Parameters Relevant for Ozone Effect	References
Preclinical trial in rats. Ozone oxidative preconditioning: A protection against cellular damage by free radicals.	Six animal groups: adult female Sprague-Dawley rats, 220–250 g. Ten animals per group. Four control groups Toxicity-producing ROS: by CCl_4_ solution; ozone preconditioning: 15 O_3_ treatments as rectal insufflation (1 mg/kg), 4.5–5 mL, 50 μg/mL.	Only a few redox parameters relevant to the ozone effect: SOD in u/g; GSH in mmol/g: TBARS in nmol/g protein.	León et al., 1998 [2].
Preclinical trial in rats. Similar protective effect of ischemic and ozone oxidative preconditioning in liver/ischemia–reperfusion injury.	Adult male Wistar rats (250–300 g). n = 32, hepatic ischaemia I/R (n = 8): 90 min hepatic ischaemia 90 min reperfusion (n = 8). ozone preconditioning: 15 O_3_ treatments as rectal insufflation (n = 8) (1 mg/kg), 5–5.5 mL, 50 μg/mL.	MDA + 4-hydroxynonenal in mmol/g; total SH-groups in mol/mg protein.	Ajamieh et al., 2002 [3].
Preclinical trial in rats. Effects of ozone oxidative preconditioning on nitric oxide generation and cellular redox balance in a rat model of hepatic ischemia–reperfusion.	60 adult male Wistar rats (250–300 g), 10 per group. hepatic ischemia I/R 90 min ischemia, 90 min reperfusion; ozone preconditioning: 15 O_3_ treatments as rectal insufflation (1 mg/kg), 5–5.5 mL, 50 μg/mL.	GSH in μg/g tissue, MDA + 4 hydroxynonenal in mmol/g.	Ajamieh et al., 2004 [4].
Preclinical trial in rats. Role of protein synthesis in protection conferred by ozone-oxidative-preconditioning in hepatic ischemia–reperfusion.	Adult male Wistar rats (10 per group, 250–275 g) hepatic ischemia 90 min ischemia, ozone preconditioning: 15 O_3_ treatments as rectal insufflation (1 mg/kg), 5–5.5 mL, 50 μg/mL.	Mn-SOD in U/g tissue (SOD 2 mitochondrial SOD); Cu/Zn-SOD in U/g tissue (SOD 1 cytosol), MDA + 4 hydroxynonenal in mmol/g.	Ajamieh et al., 2005 [5].
Preclinical trial in rats. Ozone Therapy on Rats Submitted to Subtotal Nephrectomy: Role of Antioxidant System.	30 female Wistar rats (180–200 g), 10 per group: 15 treatments 2.5−2.6mL, conc. 50 μg/mL rectal insufflation 1×/day, partial nephrectomy.	GSH (nmol/mg protein) TBARS in nmol/g protein.	Calunga et al., 2005 [6].
Preclinical trial in rats. Ischemia–reperfusion animal model on rat myocard after ozone oxidative preconditioning. [ Ischämie/Reperfusions-Modell am Herzen nach oxidativer Konditionierung durch Ozon].	30 male albino rats (100–150 g), 3 groups, each with n = 10, preventive ozone i.p. 2× per week for 2 months or for 3 months, conc.: 4 μg/mL; 28 μg per 100 g rat (400 μg per 100 mL blood), followed by 30 min ischemia and 30 min reperfusion.	SOD, MDA, Mitochondrial biogenesis.	Barakat et al., 2006 [13].
Preclinical trial in rats. Ozone Therapy Protects Against Rejection in a Lung Transplantation Model: A New Treatment?	Male Sprague-Dawley rats, n = 36, rectal O_3_ daily for 2 weeks prior to lung transplantation (20–50 μg per animal) and 50 μg/dose 3×/week up to 3 months.	Ozone pre- and postconditioning significantly decreased the expression of all genes related to oxidative stress and chronic rejection.	Santana-Rodríguez et al., 2017 [8].
Preclinical trial in rats. Ozone protects rat heart against ischemia–reperfusion injury: A role for oxidative preconditioning in attenuating mitochondrial injury.	Adult male Sprague-Dawley rats (200–250 g) OzoneOP 2 mL ozone 100 μg/kg/day for 5 days, 30 min of cardiac ischemia followed by 2 h reperfusion.	SOD 1 and SOD 2 in u/mg protein	Meng et al., 2017 [14].
Preclinical trial in cell culture. Ozone alleviates ischemia–reperfusion injury by inhibiting mitochondrion-mediated apoptosis pathway in SH-SY5Y cells.	SH-SY5Y cells as model for neuronal function tests; ozone oxidative preconditioning via incubation with 40 μg/mL and cultured for 2, 6; 12, and 24 h.	SOD in μ/mg protein, MDA in nmol/mg protein.	Cai et al., 2020 [15].
Preclinical trial in rabbits. Ozone preconditioning protects rabbit heart against global ischemia–reperfusion injury in vitro by up-regulating HIF-1.	Adult rabbits (2.50–2.75 kg), ozone oxidative preconditioning: i.p. injections 10 mL daily for 5 days. Three concs: 20; 40, 80 μg/mL followed by 20 min ischemia, 60 min reperfusion.	SOD in μ/mg protein, MDA in mmol/mg protein	Wang et al., 2022 [16].
Preclinical trial in cell culture and in an animal model (mice). Ozone pretreatment alleviates ischemia–reperfusion injury-induced myocardial ferroptosis by activating the Nrf2/Slc7a11/Gpx4 axis.	1. H9c2 cardiomyocytes; 2. Male C57 mice, 7 weeks of age, 22–24 g. n = 36, 12 per group, ozone preconditioning, 25 μg/mL i.p. injections, 2 mL per day for 5 days. 30 min. ischemia, 2 hrs reperfusion.	SOD activity in % of CTL (cytotoxic T lymphocyte activity), MDA in μmol/g protein	Ding et al., 2023 [17].
Preclinical trial in rats. Ozone-mediated cerebral protection: Unraveling the mechanism through ferroptosis and the NRF2/SLC7A11/GPX4 signaling pathway.	Sprague-Dawley (SD) rats (260–300 g), middle artery occlusion for 120 min, followed by surgery and reperfusion. Ozone oxidative preconditioning, i.p. injections (intra peritoneal), 20 μg/mL with 2.5 mL/kg/d, 5 days.	GSH in mg/g protein, LPO (lipoperoxides), MDA in μmol/g protein.	Zhu et al., 2024 [10].

## Data Availability

Data contained within the article.

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
