# Peer review of "Protection of Mitochondria, Cells and Organs from Ischemia–Reperfusion Damage Through Preventive Redox Bioregulation by Ozone"

_ijms, 2025, doi:10.3390/ijms26125557_

Round 1
Reviewer 1 Report
Comments and Suggestions for Authors
This paper "Protection of Mitochondria, Cells and Organs from
Ischemia/Reperfusion Damage Through Preventive Redox
Bioregulation by Ozone" deals with a very interesting and, from a research perspective, very popular topic. The literature review is very well done, with enough recent literature citations. As such, the paper can be accepted for publication after revision.
The authors provide a scientific review of the mechanism of action, however, a review of the number of articles on this topic is missing. Graphically and through discussion in the introductory part, the authors must show the importance of the topic. Graphically show the number of papers in the last few years in this area.
The quality of the figures is poor. The axes are poorly visible. Improve this part in the revised version.
A better critical review is missing for sections 2.3, 3.1 and 3.2. They are listed here without critical opinion.
Section: Conclusion and Future Perspective. It is not clear what future directions of research are. How the authors see progress in this field in the future. This section needs to be significantly improved.
Author Response
Response to reviewer 1
Rev. 1.
The type oft he paper will be changed from review to „communication“ on request of the rditors.
1. Our intention is to prove the mechanism of action of low dose ozone application, the so callled
ozone oxidative preconditioning, providing a good antioxidant status as defense strategy against high
oxidative stress as a consequence of I/R damage. For that purpose we need measurements of the
specific paramaters. So we had to reduce the selection of papers to those where these parameters
had been measured as listed in Table 1. 3rd column “Antioxidants / oxidative stress parameters
relevant for ozone effect”.
Before 2017 there was one main group (León et al.) persuing this ideas. an increasing number of
preclinical studies with promising results followed; all of them (12) are discussed in the paper, 8 of
them also in Figure 1. Figure 1 is a short overview of the main redox results. Upregulating the cellular
and mitochondrial
Antioxidants: Oxidative stress decreases and the redox balance is restored.
2. The figures have been revised and the former ones in the manuscript replaced.
3. The missing opinion in 2.3, 3.1 and 3.2 has been added and the perspective extended.
Changes in the text are marked in red

Reviewer 2 Report
Comments and Suggestions for Authors
I thank the authors for such a nice manuscript. In fact, ischemia-related injuries are a big problem nowadays, especially in the surgical field. In this review, the authors discuss ozone-induced damage and support their review with diagrams. I have no questions or concerns.
The authors question the mitochondrial injuries induced by high oxidative stress in this review. They describe specifically the effect of high and low ozone oxidative stress and define ways for protection and restoring the redox balance.
The topic is not completely novel, since the same authors published a similar article before (https://doi.org/10.3390/molecules29122738); however, in the article submitted to the IJMS, they focused on the ozone-induced oxidative stress in the I/R rat model. The review could help researchers investigating this point. Additionally, they explained the mechanism of ozone oxidative effect in detail, which could be used to develop therapies in these conditions. Furthermore, they describe the mitochondrial biogenesis closely related issue in these injuries, which was previously involved in the pathogenesis of SARS-CoV-2 infections (doi: 10.3390/biomedicines10092258).
Even though the review is written well; however, I have a few concerns that I accidentally overlooked in my previous report. The paragraph from line 100 to 122 looks as if it was written with an AI tool. The way of adjusting the text and points is weird. Above all, the description mentioned in this point is very short. The authors need to expand it. The same situation is also for the paragraph from line 123 to line 162. Also, in Figure 4, I cannot find the significance of the figure, although it is mentioned in the ligand!! A general comment from me is that the authors need to clarify exactly what they mean by sham? Are these the animals that are not treated at all or treated with a vehicle? They need to specify the kind of treatment used for this group. Table 1 needs some more adjustments to be used. I find the colors used for the frame are somehow distracting. The item “Subject, type of study” is confusing. Why did the authors mention “preclinical trial in rats” in many positions in the table?
The conclusion of the review fits with its purpose. To improve it, the authors need to mention exactly why the ozone-induced oxidative stress is clinically so important. The authors need to talk more about the anti-oxidative therapies involved. Additionally, they need to expand the future intended plans or therapies that their review aims at.
Round 2
Reviewer 1 Report
Comments and Suggestions for Authors
A revised version can be accepted in its form.